**Data Availability Statement:** The data are held in a public repository, and can be accessed at: https://cramsurvey.org.

**Funding:** The authors received no specific funding for this work.

# Job loss and mental health during the COVID-19 lockdown: Evidence from South Africa

**Dorrit Posel**📷*, **Adeola Oyenubi**📷, **Umakrishnan Kollamparambil**

School of Economics and Finance, University of the Witwatersrand, Johannesburg, South Africa

* dorrit.posel@wits.ac.za

## Abstract

### Objectives

Existing literature on how employment loss affects depression has struggled to address potential endogeneity bias caused by reverse causality. The COVID-19 pandemic offers a unique natural experiment because the source of unemployment is very likely to be exogenous to the individual. This study assessed the effect of job loss and job furlough on the mental health of individuals in South Africa during the COVID-19 pandemic.

### Data and methods

The data for the study came from the first and second waves of the national survey, the National Income Dynamics-Coronavirus Rapid Mobile Survey (NIDS-CRAM), conducted during May-June and July-August 2020, respectively. The sample for NIDS-CRAM was drawn from an earlier national survey, conducted in 2017, which had collected data on mental health. Questions on depressive symptoms during the lockdown were asked in Wave 2 of NIDS-CRAM, using a 2-question version of the Patient Health Questionnaire (PHQ-2). The PHQ-2 responses (0–6 on the discrete scale) were regrouped into four categories making the ordered logit regression model the most suited for assessing the impact of employment status on depressive symptoms.

### Results

The study revealed that adults who retained paid employment during the COVID-19 lockdown had significantly lower depression scores than adults who lost employment. The benefits of employment also accumulated over time, underscoring the effect of unemployment duration on mental health. The analysis revealed no mental health benefits to being furloughed (on unpaid leave), but paid leave had a strong and significant positive effect on the mental health of adults.

### Conclusions

The economic fallout of the COVID-19 pandemic resulted in unprecedented job losses, which impaired mental wellbeing significantly. Health policy responses to the crisis therefore need to focus on both physical and mental health interventions.

**Competing interests:** The authors have declared that no competing interests exist.

## Introduction

It is well documented that the COVID-19 pandemic has resulted in large increases in unemployment in many countries [1]. South Africa is no exception: studies estimate that between 2.2 and 2.8 million adults in the country lost their jobs from February to April 2020, following the lockdown and the wide-scale suspension of economic activity [2–4]. This loss of employment had significant implications for people's access to economic resources [4, 5]; and it may also be an important reason for why elevated depressive symptoms were reported among adults during the first months of the pandemic [6].

It is increasingly being recognized that the health costs of COVID-19 are not limited to physical health but include the effects of the pandemic on the individual's mental or psychological well-being [7–10]. This study explores how job loss affects people's mental health using longitudinal micro-data collected after the introduction of the COVID-19 lockdown in South Africa.

The COVID-19 pandemic offers a unique opportunity to analyze the implications of job loss for mental health, because the source of unemployment is very likely to have been exogenous to (or beyond the control of) the individual. There is a large literature which investigates how the loss of employment affects depression or anxiety, where studies compare the mental health of the employed and the unemployed [11–14]. However, testing the relationship between unemployment and depression typically is complicated by methodological problems associated with causality, which arise even with longitudinal data. This is because it is often not possible to establish the temporal ordering of events: are changes in depressive symptoms caused by, or do they precede, changes in activity status? For example, people who experience job loss may exhibit more depressive symptoms because of their unemployment; but it is also possible that those who are depressed are significantly less likely to search for, or maintain, employment [15–17].

The national lockdown in response to the COVID-19 pandemic, and the associated loss of employment, provide a natural experiment that removes these problems of causality. In addition, the labor market implications of the COVID-19 lockdown are unique because most economic activity was suspended in anticipation that (at least some) activity would resume once the lockdown was eased. Some workers therefore retained jobs to return to, but for the duration of the lockdown, they were neither working nor earning an income. For example, among adults who reported being employed during South Africa's lockdown, a sizeable share (approximately 17 percent in April) also reported that they were currently not working any hours and had not received payment, but that they had a job to return to. Of these furloughed workers, half were back at work by June, but nearly 40 percent fell into unemployment [2].

These unusual characteristics of the COVID-19 crisis make it possible to distinguish between job loss and job furlough when investigating the implications of activity status for mental health. This is an interesting distinction to draw because it offers insight into whether expectations of a job in the future provide psychological protection against the loss of current earnings and work activity.

South Africa is also an important country in which to explore the effects of job loss on mental health. There have been many decades of research, particularly in developed countries, on the psychological implications of unemployment [12, 14, 18–20]. However, although South Africa has had persistently high rates of unemployment since the transition to democracy [21, 22], there are few studies which interrogate how this unemployment affects levels of depression and anxiety in the population [23].

Existing research that assesses psychological health during the COVID-19 pandemic has relied on cross-sectional data that have been collected through online questionnaires, biasing

samples against people with limited access to the internet [8]. This type of selection bias is likely to be particularly important in developing countries such as South Africa, where access to the internet varies significantly and systematically by socio-economic status and location [24, 25].

In this study, we analyze unique longitudinal data from two waves of a rapid mobile survey, where participants were drawn from a nationally stratified sample, and information was collected using computer-assisted telephonic interviews. We use these data to investigate the extent to which job loss undermined the mental health of adults who were employed before the COVID-19 lockdown, if this effect was compounded as unemployment persisted, and whether job furlough provided any protection against the distress caused by losing a job altogether.

## Data and methods

### Data source

The data for the study come from the National Income Dynamics-Coronavirus Rapid Mobile Survey (NIDS-CRAM). NIDS-CRAM was developed by a consortium of more than 30 academics (of which one author was part), from universities across South Africa. It was introduced to track the socio-economic and health effects of the COVID-19 pandemic and the associated lockdown. It is expected that the survey will span one year, by which time, five waves will have been conducted [26]. By October 2020, two waves of NIDS-CRAM had been completed. Ethical clearance for the study was obtained from the University of Cape Town Commerce Ethics Committee (REC 2020/04/2017), with reciprocal ethics from the University of Stellenbosch. The data, which are in the public domain, are available at: https://cramsurvey. org.

To obtain a sample that was as nationally representative as possible under the circumstances, participants for NIDS-CRAM were drawn from South Africa's national household survey, the National Income Dynamics Study (NIDS). NIDS was conducted by the Southern African Labour and Development Research Unit, and the last wave was undertaken in 2017. The NIDS-CRAM sample was selected from the 2017 national sample using a stratified design but with 'batch sampling'. Sampling in batches offered flexibility in adjusting the sample rate as the surveying progressed, and as information about stratum response became available [27].

The first wave of NIDS-CRAM, which was conducted from 7 May to 27 June 2020, surveyed 7073 adults aged 18 years and older. In the second wave, which was undertaken from 13 July to 13 August 2020, 5676 adults were successfully re-interviewed, yielding a response rate of 80.2 percent [28]. Attrition from Wave 1 to Wave 2 of NIDS-CRAM is estimated to be random based on observed covariates, when measured using goodness-of-fit statistics [28]. A test of attrition using probit models [29] also shows that there is no relationship between mental health and the probability of not being interviewed in NIDS-CRAM Wave 1, or of not remaining in the sample from Wave 1 to Wave 2.

All interviews for NIDS-CRAM have been conducted telephonically by call-center agents, and the instrument has been designed to take no longer than twenty minutes per interview [26]. Consequently, the questionnaire is far shorter than typical household questionnaires undertaken in South Africa, including the instrument for NIDS 2017. The Wave 1 questionnaire was translated into 10 of the 11 official languages in South Africa, while the Wave 2 questionnaire was conducted in all 11 languages.

All participants in NIDS-CRAM were informed verbally before they were interviewed that participation in the study was voluntary, and that their participation could be stopped at any time. Consent and the telephonic interview were recorded, but participants were advised that

all information collected would be kept confidential and that the information released in the datasets would be anonymized.

## Depressive symptoms

In order to increase the scope of information collected in short interviews, not all modules in the NIDS-CRAM questionnaire are repeated across waves. Of interest to this study are the questions on mental health, which were included in the Wave 2 questionnaire, but not in Wave 1. However, information on mental health was also collected in NIDS 2017.

NIDS 2017 included the ten questions which make up the Center for Epidemiologic Studies Short Depression Scale (CES-D 10). Individuals were asked about their emotional health over the past week, including whether they felt "hopeful", "fearful" "lonely" and "happy". In the far shorter questionnaire for NIDS-CRAM Wave 2, information on depressive symptoms was collected using a 2-question version of the Patient Health Questionnaire (PHQ-2) [6]. Respondents were asked whether over the previous two weeks, they "had little interest or pleasure in doing things" (question G11); and whether they had "been feeling down, depressed or hopeless" (question G12). Response options included "not at all", "several days", "more than half the days" and "nearly every day" (which we have coded from 0 to 3). The PHQ-2 is a shortened version of the widely used PHQ-9 [24], and both the PHQ-9 and the CES-D 10 have been validated as reliable screening measures of depression, including for South Africa [30].

Given differences in the information collected, measures of mental health in NIDS and NIDS-CRAM are not directly comparable. The study is therefore unable to use individual fixed effects models (or intra-individual comparisons) to control for any unobserved time-invariant factors (such as personality) that influence both depressive symptoms and activity status. It is also not possible to draw robust conclusions about how mental health has changed from 2017 (pre-COVID) to 2020 (COVID). However, the CESD-D 10 scores from 2017 are included as a covariate in the multivariate regression analysis of depressive symptoms in 2020, to offer some control both for variation in the individual propensity to exhibit depressive symptoms [23] and for possible anchoring effects in how respondents assess their symptoms [31].

The PHQ-2 scale ranges from 0 to 6, and the CES-D 10 scale, from 0 to 30, with both increasing in depressive symptoms. Both scales are employed as a continuum of distress [23, 32–34], rather than imposing a threshold to identify depression, because the appropriate cut-off has been found to vary across different language groups in South Africa [30].

## Sample and variables

The focus of the study is on the relationship between employment status and mental health during COVID-19. The first wave of NIDS-CRAM established whether adults had been working in February, prior to the start of the 'hard' lockdown in South Africa (referred to as alert level 5) when all non-essential economic activity was suspended. Detailed information was also collected on whether adults had been working in April, the number of hours worked in a typical week and whether (and what) earnings had been received. The second wave of NIDS-CRAM collected information on labor market activity in June, by which time South Africa had progressed to alert level 3 of the lockdown, and many businesses were able to re-open.

The sample for the study is all adults who were employed in the month before the COVID-lockdown started. Of these 3408 adults, 2213 were interviewed in both Waves 1 and 2 of NIDS-CRAM and have complete (non-missing) information for all the main variables included in the study. The study does not use survey weights to generate population estimates partly because the available weights are benchmarked to a sample in 2017, which as a fifth wave of

the NIDS panel, was itself not nationally representative. Further, our sample is restricted to those who were employed before the lockdown, and the weights are not stratified by employment status. We therefore consider a model-based approach more suitable [35], and we refer to our estimates as sample estimates.

For the empirical analysis, we first identify adults who reported having a job in April and a job in June. Although the time span is short, distinguishing the two periods may shed light on whether the negative effects of job loss are compounded as the duration of joblessness increases [36]. We then differentiate *among the employed* in April and June, identifying: adults who were working and earning a non-zero income; adults who were not working but still earning an income (and therefore most likely on paid leave); and adults who were neither working nor earning an income but who identified that they had a job to return to (whom we refer to as furloughed).

The multivariate analysis also includes a range of variables that are commonly adopted in empirical studies of depression, and which may moderate the relationship between activity status and depressive symptoms [23, 33, 34]. These are first, the adult's demographic characteristics: age and age squared; sex (female); marital status (partnered); educational attainment (tertiary education); race (African, where the omitted category, non-African, includes the three other race categories always identified in South African surveys, viz., Colored (of mixed race), Indian (of Asian descent) and white); and whether the individual has a chronic health condition. We also control for the adult's geographical location (urban); the type of dwelling (formal dwelling such as a house or a flat, informal dwelling or a shack, with a traditional dwelling as the omitted category); and household composition (living in a household with children aged 17 or younger). To avoid endogeneity between employment status and household income, socio-economic status is captured with information collected in NIDS on the adult's net worth in 2017, and whether at least one child support grant or older persons grant (the two most common social grants in South Africa) was received in the household in April and then in June 2020. Finally, we identify people's attitudes to COVID-19 with a binary variable for whether the respondent believed that it was possible to avoid being infected by the coronavirus.

## Statistical analysis

As the PHQ-2 is measured on a 0–6 discrete scale, we modelled the impact of employment status on depressive symptoms using the ordered logit model. The standard assumption is that there exists a latent index $y^*$ that depends linearly on a set of covariates i.e.

$$y_i^* = x_i'\beta + \varepsilon_i \tag{1}$$

where $\beta$ is a vector of parameters, $x$ represents the covariates and $\varepsilon_i$, is the error term, which is assumed to be independent and identically distributed. The observed rating of the depression score depends on the value of the latent $y_i^*$. What is observed can therefore be described as

$$
\begin{aligned}
Y_i &= 0 \ if \ y_i^* \leq 0 \\
Y_i &= 1 \ if \ 0 < y_i^* \leq \mu_1 \\
Y_i &= 2 \ if \ \mu_1 < y_i^* \leq \mu_2 \\
&\vdots \\
Y_i &= 5 \ if \ \mu_4 < y_i^* \leq \mu_5 \\
Y_i &= 6 \ y^* > \mu_5
\end{aligned}
\tag{2}
$$

where $\mu_j$ are unknown threshold parameters that are estimated. In the ordered logit model, the

estimated coefficients ($\beta$) provide information about whether the odds of being in a particular category are positive or negative, but they do not describe the magnitude of the depression score change for a unit change in $x$. We therefore also report the marginal effects which identify the effect of differences in $x$ on the probability of being in a particular category of the PHQ-2 scale.

A key assumption underlying the ordered logit regression is the proportional odds or parallel regression assumption, viz. that the same relationship exists between all the categories of the ordinal scale. The assumption is tested using the Stata post-estimation command 'oparallel' that compares the ordered logit model with a full generalized ordered logit model, which relaxes the parallel regression assumption on all explanatory variables. The null hypothesis, that there is no difference in the coefficients between models, is tested using the Wald test, Wolfe-Gould test, Likelihood ratio test, Brant test and Score test [37]. An insignificant outcome indicates that there is not enough strong evidence against the parallel regression assumption.

The ordered logit regressions with the PHQ-2 scale from 0 to 6 violated the parallel regression assumption. We therefore regrouped the scale into four categories: 0; 1 (1 or 2 of the original scale); 2 (3 or 4 of the original scale); and 3 (5 or 6 of the original scale). These regressions satisfied the parallel regression assumption and the estimated coefficients remained robust for both the original scale and the regrouped scale. (The main marginal effects from the regressions with the original scale, and the tests of the parallel regression assumption, are reported in the Tables 6–9 in S1 Appendix.)

**Results.** Among the sample of adults who were employed before the implementation of the hard lockdown in South Africa (Table 1), the modal PHQ-2 score was 0 (respondents had not experienced any depressive symptoms in the previous 2 weeks), accounting for 47% of adults. However, if a PHQ-2 score of 3 or larger is taken as the cut-off for depression [38], then almost a quarter (24%) of adults in the sample would be classified as depressed. If a CES-D 10 score of 10 or more is considered indicative of depression [33], then among this same group of adults, 17% were depressed in 2017.

In the first month following the lockdown, 30% of adults had lost their jobs, while a further 12% were furloughed. Only 41% of all adults who had been employed before the COVID-19 crisis were still actively working and earning an income, and 17% were on paid leave. Two months later, after the lockdown conditions had eased, the share of adults who were actively working had increased to 57%, and only 6% were on paid leave. The percentage of adults who were furloughed also dropped to 5%, but the share who was unemployed increased slightly to 32%.

Compared to adults who lost their job over the lockdown period, PHQ-2 scores were significantly lower among adults who retained employment (Tables 2 and 3). Moreover, the protection from depression associated with employment, or the risk of depression among those who lost their jobs, was compounded over time. Adults who retained their jobs in Wave 1 were 5.1% more likely than those who did not have jobs to report no depressive symptoms (Regression 1, Table 3) and a further 6% more likely if they also retained their job in Wave 2 (Regression 2, Table 3).

However, the employed were not all equally protected against adverse mental health. There is no significant relationship between PHQ-2 scores and being furloughed. Adults who were neither working any hours nor earning any income were therefore no more likely than adults who had lost their job to have low PHQ-2 scores on average, even if they reported having a job to return to (Tables 2 and 3).

In each wave, adults who had been actively working were 5–6% more likely to report no depressive symptoms than those who had lost employment. There was at most a weak negative

**Table 1. Descriptive statistics of adults who were employed before the lockdown.**

| Variable | Mean | Standard deviation | Minimum | Maximum | Sample |
|---|---|---|---|---|---|
| Outcome variable | | | | | |
| PHQ (0) | 0.469 | (0.499) | 0 | 1 | 2213 |
| PHQ (1) | 0.151 | (0.358) | 0 | 1 | 2213 |
| PHQ (2) | 0.142 | (0.349) | 0 | 1 | 2213 |
| PHQ (3) | 0.131 | (0.338) | 0 | 1 | 2213 |
| PHQ (4) | 0.053 | (0.225) | 0 | 1 | 2213 |
| PHQ (5) | 0.014 | (0.118) | 0 | 1 | 2213 |
| PHQ (6) | 0.039 | (0.194) | 0 | 1 | 2213 |
| Independent variables | | | | | |
| CES-D 10 score (2017) | 6.39 | (4.317) | 0 | 26 | 2213 |
| Working in wave 1 | 0.413 | (0.492) | 0 | 1 | 2213 |
| Paid leave in wave 1 | 0.173 | (0.378) | 0 | 1 | 2213 |
| Furloughed in wave 1 | 0.116 | (0.320) | 0 | 1 | 2213 |
| Not employed in wave 1* | 0.296 | (0.457) | 0 | 1 | 2213 |
| Working in wave 2 | 0.565 | (0.496) | 0 | 1 | 2213 |
| Paid leave in wave 2 | 0.063 | (0.243) | 0 | 1 | 2213 |
| Furloughed in wave 2 | 0.047 | (0.212) | 0 | 1 | 2213 |
| Not employed in wave 2* | 0.324 | (0.468) | 0 | 1 | 2213 |
| Age | 39.152 | (11.54) | 18 | 89 | 2213 |
| Female | 0.577 | (0.494) | 0 | 1 | 2213 |
| African | 0.829 | (0.376) | 0 | 1 | 2213 |
| Tertiary education | 0.276 | (0.447) | 0 | 1 | 2213 |
| Partnered | 0.493 | (0.500) | 0 | 1 | 2213 |
| Chronic health condition | 0.216 | (0.411) | 0 | 1 | 2213 |
| Urban area | 0.736 | (0.441) | 0 | 1 | 2213 |
| Formal dwelling | 0.781 | (0.414) | 0 | 1 | 2213 |
| Informal dwelling (shack) | 0.112 | (0.316) | 0 | 1 | 2213 |
| Traditional dwelling (mud)* | 0.107 | (0.309) | 0 | 1 | 2213 |
| Living with children | 0.753 | (0.431) | 0 | 1 | 2213 |
| Social grant $\geq$ 1 (wave 1) | 0.656 | (0.475) | 0 | 1 | 2213 |
| Social grant $\geq$ 1 (wave 2) | 0.646 | (0.478) | 0 | 1 | 2213 |
| Log (individual net worth) 2017 | 10.183 | (1.909) | 2.303 | 17.858 | 1941 |
| Coronavirus can be avoided | 0.886 | (0.318) | 0 | 1 | 1941 |

Source: NIDS 2017; NIDS-CRAM waves 1 and 2.

Notes

* Reference category in the regressions

relationship between having had paid leave in Wave 1, and depression scores in Wave 2. But adults who were on paid leave in the wave that depression scores were collected reported significantly lower scores, even compared to adults who were actively working in that month ($\chi^2$ = 8.02, p < 0.02). Adults on paid leave in Wave 2 were also 10% less likely than adults who had lost their job to report no depressive symptoms (Table 4).

These results remain robust when the set of control variables is expanded to include a measure of the individual's net wealth (three years prior) and their assessment of whether contracting the coronavirus can be avoided (although the sample size was considerably reduced because of large numbers of non-response to these questions) (Regression 4 Tables 2 and 5).

**Table 2. Ordered logit regressions predicting PHQ-2, among those employed before lockdown.**

| | (1) | (2) | (3) | (4) |
|---|---|---|---|---|
| | Regression | Regression | Regression | Regression |
| CES-D 10 score (2017) | 0.005 | 0.005 | 0.005 | 0.005 |
| | (0.010) | (0.010) | (0.010) | (0.010) |
| Employed (W1) | -0.203** | -0.227** | | |
| | (0.096) | (0.103) | | |
| Employed (W2) | -0.239** | -0.225** | | |
| | (0.097) | (0.104) | | |
| Working (W1) | | | -0.198* | -0.239** |
| | | | (0.109) | (0.117) |
| Paid leave (W1) | | | -0.190 | -0.240* |
| | | | (0.128) | (0.138) |
| Furlough (W1) | | | -0.128 | -0.128 |
| | | | (0.141) | (0.149) |
| Working (W2) | | | -0.256** | -0.221** |
| | | | (0.103) | (0.110) |
| Paid leave (W2) | | | -0.398** | -0.439** |
| | | | (0.184) | (0.198) |
| Furlough (W2) | | | 0.047 | 0.021 |
| | | | (0.201) | (0.222) |
| Age | 0.028 | 0.025 | 0.029 | 0.026 |
| | (0.020) | (0.021) | (0.020) | (0.021) |
| Age$^2$/100 | -0.039* | -0.035 | -0.040* | -0.036 |
| | (0.022) | (0.024) | (0.022) | (0.024) |
| Female | - 0.029 | - 0.060 | - 0.024 | - 0.054 |
| | (0.085) | (0.092) | (0.086) | (0.092) |
| African | -0.737*** | -0.782*** | -0.747*** | -0.789*** |
| | (0.112) | (0.122) | (0.113) | (0.123) |
| Partnered | 0.053 | 0.046 | 0.056 | 0.049 |
| | (0.086) | (0.092) | (0.086) | (0.093) |
| Tertiary education | 0.135 | 0.105 | 0.138 | 0.109 |
| | (0.093) | (0.103) | (0.094) | (0.103) |
| Urban | 0.283*** | 0.268*** | 0.286*** | 0.271*** |
| | (0.097) | (0.103) | (0.097) | (0.103) |
| Living with children | 0.029 | 0.046 | 0.033 | 0.050 |
| | (0.109) | (0.117) | (0.109) | (0.118) |
| Formal dwelling | 0.151 | 0.176 | 0.156 | 0.184 |
| | (0.133) | (0.141) | (0.133) | (0.141) |
| Informal dwelling (shack) | -0.023 | -0.091 | -0.016 | -0.079 |
| | (0.177) | (0.190) | (0.177) | (0.190) |
| Chronic health condition | 0.187* | 0.243** | 0.191* | 0.251** |
| | (0.101) | (0.108) | (0.101) | (0.108) |
| Social grant ≥ 1 (wave 1) | 0.173 | 0.143 | 0.183 | 0.149 |
| | (0.131) | (0.139) | (0.131) | (0.139) |
| Social grant ≥ 1 (wave 2) | -0.225* | -0.230 | -0.233* | -0.237* |
| | (0.135) | (0.143) | (0.135) | (0.143) |
| Coronavirus avoided | | -0.208 | | -0.203 |
| | | (0.135) | | (0.136) |

*(Continued)*

**Table 2.** (Continued)

| | (1) | (2) | (3) | (4) |
|---|---|---|---|---|
| | Regression | Regression | Regression | Regression |
| Log (wealth) (2017) | | 0.009 | | 0.009 |
| | | (0.028) | | (0.028) |
| /cut1 | -0.147 | -0.307 | -0.117 | -0.265 |
| | (0.448) | (0.515) | (0.449) | (0.517) |
| /cut2 | 1.191*** | 1.005* | 1.222*** | 1.048** |
| | (0.448) | (0.516) | (0.449) | (0.517) |
| /cut3 | 2.935*** | 2.743*** | 2.966*** | 2.787*** |
| | (0.455) | (0.522) | (0.456) | (0.524) |
| Observations | 2,213 | 1,941 | 2,213 | 1,941 |

Standard errors in parentheses

*** p<0.01

** p<0.05

* p<0.1

As the lagged depression score from 2017 is measured using a different instrument and therefore captures depressive symptoms on a more extensive scale, we also tested the robustness of the findings to alternative specifications. First, we converted both the PHQ-2 and CES-D 10 scores to binary variables using the threshold that is often adopted in studies from other countries (a score of 3 or higher for the PHQ-2 and of 10 or more for the CES-D) and estimated logit regressions. Second, we ran the ordered logit regressions without the depression score from 2017; and third, we normalized both the PHQ-2 and the CES-D 10 scores and estimated ordinary least squares regressions. The results from these tests are reported in the Tables 10a-10c in S1 Appendix. Overall, the findings are consistent with the original estimations and all variables retain significance in the latter two sets of regressions, although some of the activity status variables lose significance in the binary specification.

**Table 3. Marginal effects (Regressions 1 and 2).**

| | Regression (1) | | Regression (2) | |
|---|---|---|---|---|
| VARIABLES | Employed W1 | Employed W2 | Employed W1 | Employed W2 |
| PHQ-2(0) | 0.051** | 0.060** | 0.057** | 0.056** |
| | (0.024) | (0.024) | (0.026) | (0.026) |
| PHQ-2(1,2) | -0.015** | -0.017** | -0.016** | -0.016** |
| | (0.007) | (0.007) | (0.008) | (0.008) |
| PHQ-2(3,4) | -0.026** | -0.031** | -0.029** | -0.029** |
| | (0.012) | (0.013) | (0.013) | (0.013) |
| PHQ-2(5,6) | -0.010** | -0.011** | -0.011** | -0.011** |
| | (0.005) | (0.005) | (0.005) | (0.005) |
| n | 2,213 | 2,213 | 1,941 | 1,941 |

Standard errors in parentheses

*** p<0.01

** p<0.05

* p<0.1

**Table 4. Marginal effects (Regression 3).**

| VARIABLES | Regression (3) | | | | | |
|---|---|---|---|---|---|---|
| | Working W1 | Paid leave W1 | Furlough W1 | Working W2 | Paid leave W2 | Furlough W2 |
| PHQ-2(0) | 0.049* | 0.047 | 0.032 | 0.064** | 0.099** | -0.012 |
| | (0.027) | (0.032) | (0.035) | (0.026) | (0.046) | (0.050) |
| PHQ-2(1,2) | -0.014* | -0.014 | -0.009 | -0.018** | -0.029** | 0.003 |
| | (0.008) | (0.009) | (0.010) | (0.008) | (0.013) | (0.014) |
| PHQ-2(3,4) | -0.026* | -0.025 | -0.017 | -0.033** | -0.052** | 0.006 |
| | (0.014) | (0.017) | (0.018) | (0.013) | (0.024) | (0.026) |
| PHQ-2(5,6) | -0.009* | -0.009 | -0.006 | -0.012** | -0.019** | 0.002 |
| | (0.005) | (0.006) | (0.007) | (0.005) | (0.009) | (0.009) |
| n | 2,213 | 2,213 | 2,213 | 2,213 | 2,213 | 2,213 |

Standard errors in parentheses

*** $p < 0.01$

** $p < 0.05$

* $p < 0.1$

## Discussion

Although employment is typically far less secure in developing countries, there has been little research on the relationship between mental health, employment and joblessness in these countries [14], with studies focusing more on the association between mental health and poverty [17, 39]. In South Africa, there is a growing body of empirical literature which has estimated the correlates of depression or depressive symptoms [15, 23, 33, 34, 40]; but despite South Africa's very high unemployment rate, there is no work that has specifically explored how job loss, or the lack of employment, affects an adult's vulnerability to depression.

This study analysed longitudinal micro-data collected in 2020, during the COVID-19 lockdown in South Africa, from a sample of adults who had been previously interviewed in a national household survey in 2017. The analysis was restricted to adults who were employed shortly before the introduction of the hard lockdown and the subsequent wide-spread loss of

**Table 5. Marginal effects (Regression 4).**

| VARIABLES | Regression (4) | | | | | |
|---|---|---|---|---|---|---|
| | Working W1 | Paid leave W1 | Furlough W1 | Working W2 | Paid leave W2 | Furlough W2 |
| PHQ-2(0) | 0.060** | 0.060* | 0.032 | 0.055** | 0.109** | -0.005 |
| | (0.029) | (0.034) | (0.037) | (0.027) | (0.049) | (0.055) |
| PHQ-2(1,2) | -0.017** | -0.017* | -0.009 | -0.016** | -0.032** | 0.002 |
| | (0.009) | (0.010) | (0.011) | (0.008) | (0.015) | (0.016) |
| PHQ-2(3,4) | -0.031** | -0.031* | -0.017 | -0.029** | -0.057** | 0.003 |
| | (0.015) | (0.018) | (0.019) | (0.014) | (0.026) | (0.029) |
| PHQ-2(5,6) | -0.011** | -0.011* | -0.006 | -0.010** | -0.021** | 0.001 |
| | (0.006) | (0.007) | (0.007) | (0.005) | (0.009) | (0.010) |
| n | 1,941 | 1,941 | 1,941 | 1,941 | 1,941 | 1,941 |

Standard errors in parentheses

*** $p < 0.01$

** $p < 0.05$

* $p < 0.1$

employment. Although employment started to recover as the lockdown conditions eased, corresponding to the second wave of the data collected, adults remained considerably less likely to be employed than before the lockdown started.

We used ordered logit models to investigate the relationship between depressive symptoms and job loss during the COVID-19 crisis. As the source of job loss following the nation-wide lockdown was exogenous to the individual, the relationship between depression scores and activity status was not biased by selection issues; viz. that individuals with poor mental health were more likely to lose their jobs. In addition, prior depression scores (the adult's CES-D score from the 2017 data) were included as a covariate in the regression models, to control for unobserved differences in personality or genetic endowments, which may have affected not only vulnerability to depression but also how symptoms were recalled and reported.

Consistent with what would be expected from studies on unemployment and depression, adults who retained employment during the COVID-19 lockdown reported significantly lower depression scores than adults who lost employment. The benefits of employment also accumulated over time, as employment in each wave resulted in significantly lower scores. This finding is consistent with studies that show how the duration of unemployment is associated with increasing negative effects on mental health. A distinction is often drawn between short-term unemployment ($< 6$ months), and long-term unemployment ($\geq 6$ months) [14], but in this study, the trend was evident also over the first few months of unemployment.

The estimations included a historical measure of the individual's economic status (their individual net worth in 2017), but because earnings are the largest source of income in the household, household income was not included as a covariate. The association between unemployment and mental health therefore arises partly because job loss threatens the economic security of the individual (and the household) [41], and also because of the psychological trauma associated with a loss of identity, purpose and structure of time [19].

When the employed were disaggregated into three groups (actively working and earning, on paid leave, not working or earning) the analysis revealed no mental health benefits to being furloughed. Any protective effect of 'having a job to return to' was likely undermined by the loss of current income, and anxiety over when and whether work would resume. In contrast, the analysis identified strong mental health benefits of recently taken *paid* leave (in Wave 2), even if this leave was spent during times of COVID-19.

The regression analysis also suggested that social grants (or cash transfers) may provide some protection against the incidence of depressive symptoms. Social grants are an integral part of the livelihood strategies of poor households in South Africa. This is the case even in households where adults have employment, because much of this employment involves low-waged work [5]. Most adults in the study's sample lived in a household where at least one social grant for a child (the child support grant) or the elderly (the older persons grant) was received. Depression scores were lower when social grants were received but the association was only weakly significant (at the 10% level) and only for social grants received in Wave 2 i.e. there is no suggestion that any protective effects of social grants endure beyond a month. The value of social grants is insufficient to lift most households above the poverty line [42]; but the expansion of the social grant system has been associated with a large decline in the incidence of hunger reported in households [43], and the importance of social grant receipt would have been amplified during the COVID-19 crisis.

The coefficients on the other covariates included in the regression models were mostly aligned with findings from South African studies which have analyzed (pre-COVID-19) national micro-data [23, 34, 44]. Vulnerability to depression increased non-linearly with age; and it was significantly higher among adults who reported a chronic health condition and who lived in an urban area (relative to a rural area). Contrary to other studies, however, our results

consistently showed that on average, Africans reported significantly lower levels of depression than non-Africans. One possible explanation for this unexpected finding is that it reflects a "steeling effect" [45] among Africans, who likely have experienced much more past adversity than non-Africans, and who may therefore have acquired more resilience in dealing with negative events.

In the first months of the pandemic, COVID-19 was sometimes presented as the 'great equalizer', in that the people who travelled (and who therefore may have had higher socio-economic status) were initially more likely to be infected [46]. The development of the pandemic has shown that COVID-19 is not blind to socio-economic status [5, 46]; but it is also not a pandemic that is confined only to the poor or disadvantaged. Although Africans were significantly more likely than non-Africans to experience job loss during the lockdown in South Africa [2, 43], it also exposed non-Africans to far greater economic shocks, on average, than they were likely to have experienced previously. In comparison to Africans, who have suffered high rates of poverty and unemployment as a legacy of apartheid and racial exclusion, non-Africans therefore may not have developed as effective coping strategies to overcome the difficult circumstances associated with the COVID-19 crisis.

In contrast to research on the mental health implications of COVID-19 in the UK, there is no evidence that the effects of job loss on mental health were gendered (an interactive term for African and female did not yield significant results in any of the estimations) [10].

## Conclusion

The lockdown in response to the COVID-19 pandemic resulted in sizeable job losses in South Africa (and around the world). This exogenous shock provided a natural experiment to investigate how job loss affects mental health. The labor market implications of the COVID-19 lockdown were also unique because many workers retained jobs to return to, but for the duration of the lockdown, they were neither working nor earning an income.

This study showed that among a sample of adults who were employed before the lockdown in South Africa, those who lost their jobs or whose jobs were furloughed reported significantly more vulnerability to depression than those who retained employment. It is also possible that the severity of depressive symptoms has been underestimated in the PHQ-2 measures analyzed in the study. The shortened version of the Patient Health Questionnaire is an attractive measure of depression when there are stringent constraints on data collection (as has been the case during COVID-19). However, as it based on only two questions, it is less sensitive to variation in, or the severity of, depressive symptoms in contrast to more comprehensive measures such as the PHQ-9 and the CES-D 10 [47].

After HIV and other infectious disorders, mental health and nervous system disorders are the third highest contributor to the burden of disease in South Africa [48]. However, mental disorders are far less likely to be treated than physical disorders [49]. The provision of mental health services has been decentralized and moved to communities and districts hospitals; but the scale of services remains inadequate [49, 50] and mental health services in South Africa have been significantly underfunded [51]. One of the stated objectives of the South African Declaration on the Prevention and Control of Non-Communicable Diseases is to increase the number of people screened and treated for mental illness by 30 percent by 2030 [52]. The effects of the COVID-19 crisis on mental health make this objective even more salient.

Mental health interventions and support by themselves cannot solve the underlying problem of job loss as a result of a widespread event like the pandemic; but they can help the individual stay confident and motivated to persevere with job search when the economy rebounds. Apart from this, more specialized programmes that address the needs of job seekers through,

for example, retraining initiatives and skills development, including those related to job search and dealing with rejection, need to be put in place to enhance the probability of re-employment. These interventions are relevant not only in response to the COVID-19 pandemic but also more generally, in the context of South Africa's persistently high rate of unemployment.

## Supporting information

**S1 Appendix.**
(DOCX)

## Acknowledgments

The authors thank two anonymous reviewers for their helpful comments.

## Author Contributions

**Conceptualization:** Dorrit Posel.

**Formal analysis:** Dorrit Posel, Adeola Oyenubi, Umakrishnan Kollamparambil.

**Methodology:** Dorrit Posel, Adeola Oyenubi, Umakrishnan Kollamparambil.

**Writing – original draft:** Dorrit Posel.

**Writing – review & editing:** Dorrit Posel, Adeola Oyenubi, Umakrishnan Kollamparambil.

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
