## [Decision Letter · Decision Letter 0]

22 Jan 2021

PONE-D-20-38028

Job loss and mental health during the COVID-19 lockdown: Evidence from longitudinal micro-data for South Africa

PLOS ONE

Dear Dr. Posel,

Thank you for submitting your manuscript to PLOS ONE. After careful consideration, we feel that it has merit but does not fully meet PLOS ONE’s publication criteria as it currently stands. Therefore, we invite you to submit a revised version of the manuscript that addresses the points raised during the review process.

We look forward to receiving your revised manuscript.

Kind regards,

Gabriel A. Picone

Academic Editor

PLOS ONE

2. Please include a copy of Tables 4 and 5 which you refer to in your text on page 13.

Reviewers' comments:

Reviewer's Responses to Questions

**Comments to the Author**

1. Is the manuscript technically sound, and do the data support the conclusions?

Reviewer #1: Partly

Reviewer #2: Yes

2. Has the statistical analysis been performed appropriately and rigorously? 

Reviewer #1: Yes

Reviewer #2: Yes

3. Have the authors made all data underlying the findings in their manuscript fully available?

Reviewer #1: Yes

Reviewer #2: Yes

4. Is the manuscript presented in an intelligible fashion and written in standard English?

Reviewer #1: Yes

Reviewer #2: Yes

5. Review Comments to the Author

Reviewer #1: Summary

This is an interesting paper which offers an analysis of protective and risk factors of mental distress focusing on being employed or being furloughed in South Africa, after controlling for several sociodemographic factors and a measure of mental health before the pandemic. The paper is in general well written but I have both major and minor comments that need to be carefully addressed in a revision.

Major comments

(1) Contribution and related literature: Recent published work has tried to understand how changes in mental wellbeing between pre-COVID and COVID periods can be related to being employed or being furloughed (Banks and Xu, 2020): https://onlinelibrary.wiley.com/doi/full/10.1111/1475-5890.12239. The current paper must frame its methodology, limitations and findings in the context of what has already been done and is already known. How do the methodology and main findings compare with recent COVID studies on employment and mental health? For instance, how do the PHQ-2 and the GHQ-12 measures (the GHQ-12 is used in Banks and Xu, 2020) compare when measuring mental health?

(2) Selective attrition: Does mental health at baseline predict participation in W1 and W2?

(3) Comparability of measures of mental health: What do we know about the relationship between the PHQ-2 and the CES-D 10 from previous studies? What is the fraction of individuals with PHQ>=3 and CES D -10 >=10 in previous studies?

(4) Sample: The study focuses on all adults who were employed in the month before the COVID-lockdown started. Given that the focus is on employment status, it would be more appropriate to focus on a more restricted age group: 18-64. Currently, the maximum age (Table 1) is 89!

(5) Sampling weights: The authors do not mention the use of sampling weights, but they should explain why they do not use the available NIDS (-CRAM) sampling weights. If there is no reason that justifies their choice, I am afraid that sampling weights must be used.

(6) The discussion section can be broken down into two sections: “discussion” and a “conclusion”. Pros and cons of the

study must be clearly acknowledged. Cons of the study include the fact that not only mental health is measured on a different scale before the pandemic, but a different collection method is used. This should probably be discussed, if not accounted for.

(7) The finding on race/ethnicity (African vs. non-African) is intriguing. Recently, Proto and Quintana-Domeque (2020) show that in the UK there are differences by gender and ethnicity in the deterioration in mental health between pre-COVID and COVID periods. Given the previous research by Proto and Quintana-Domeque (2020) and the findings in this paper, the authors should include the interaction between gender and race. In addition, it is standard practice to define gender as female (and the reference category as male), so that the new regressions should include the following three dummies: Female, African, and Female*African.

Minor comments

a. The title of the manuscript seems a bit misleading: the authors acknowledge that they cannot control for individual fixed effects since they do not observe the dependent variable before the pandemic. This limitation seems important when presenting their evidence as longitudinal.

b. The numbering of tables is not correct. The text refers to Table 3 (p.11), but indeed, there is Table 2 and then “Table 1”, “Table 2” and Table 3” reporting the marginal effects for regressions 1-2, 3 and 4, respectively. Similarly, the text refers to Table 5 (p.13), but there is no Table 5.

c. Typos: Table 1 reports a mean of 10.184 for the binary variable “Coronavirus can be avoided”. This should be fixed.

d. Pages 9-10: Equations are not numbered. Moreover, the variables y*, x and Y should be indexed with i, and the cut-off mu should not be indexed with i, but with a different letter.

e. Page 10: The authors write “An insignificant outcome indicates that the assumption has been met.” A more precise statement follows: “An insignificant outcome indicates that there is not enough strong evidence against the parallel regression assumption.”

f. Some of the findings are statistically significant at the 10% level (e.g. Social grants): “The claim that the regression analysis pointed to the importance of social grants (or cash transfers) in providing protection against the incidence of depressive symptoms” seems too strong. The causal language should be tuned down and the authors should not emphasize statistically significant findings at the 10% level.

References

• Banks, J. and Xu, X. (2020) “The Mental Health Effects of the First Two Months of Lockdown during the COVID-19 Pandemic in the UK,” Fiscal Studies: https://onlinelibrary.wiley.com/doi/full/10.1111/1475-5890.12239

• Proto, E. and Quintana-Domeque, C. (forthcoming) “COVID-19 and mental health deterioration by ethnicity and gender in the UK”, PLOS ONE. Previous version: https://www.iza.org/publications/dp/13503/covid-19-and-mental-health-deterioration-among-bame-groups-in-the-uk

Reviewer #2: 1. Is the manuscript technically sound, and do the data support the conclusions?

The authors have gone to painstaking lengths to eradicate nearly all concerns about the causal relationship between job loss, furlough, and mental health. Starting with the basics of making a case for an exogenous relationship between job loss and mental health, to noting the sampling to minimize bias found in typically online-only sampling (for rapid sampling, the authors have done an impressive job), to drawing upon a well-established nationally representative survey, to the missingness of data and being transparent that the data appear missing at random based on observed covariates (a scientifically transparent and important distinction from truly missing at random or missing completely at random), to disclosing the inability to account for individual fixed effects pertaining to influences on mental health. I expect that the scholarly community reading this paper may take issue with the exogeneity argument, particularly as those at the margins of the workforce in South Africa—in very tenuous employment, of which many Black South Africans are in—who could have been on the verge of losing their jobs anyway. Nonetheless, this is about as good as one could get for research design and assessing cause and effect. The conclusions drawn are very appropriately rooted in the methods and data.

2. Has the statistical analysis been performed appropriately and rigorously?

In short, yes.

The CES-D 10 is appropriate and externally (and within South Africa among different racial/ethnic groups) validated for measuring depression. The regression models are soundly developed and the authors clearly have not taken any shortcuts—little things like adjusting for age by also including the quadratic are very understated but key to making this research technically sound). Further, noting the parallel regression assumption and how they adjusted for that in their programming is generally considered “best practices” and something that most scholars take for granted/do not discuss when presenting ordered logistic regression models.

I am surprised that the results are not stratified by race/ethnicity as this moderates nearly everything in South Africa. Another suggestion would be to specify your variable coding a bit more. The average reader may be confused by the African versus non-African distinction and what it means in South Africa—I too am a bit confused mainly because I would like to know which category Coloured South Africans fall into (I also recognize that Indian and White South Africans account for only a small share of the overall population). The tables seem out of order or out of place in the manuscript too. The results are all there but the table numbering is off.

*3. Have the authors made all data underlying the findings in their manuscript fully available?

I confirmed that these data are accessible through the NIDS website; they are thus publicly available for anyone, so long as a user agreement is signed.

*4. Is the manuscript presented in an intelligible fashion and written in standard English?

The paper is very well-written but I suggest an additional round of copy-editing and sorting out the issue with table numbering.

6. PLOS authors have the option to publish the peer review history of their article (what does this mean?). If published, this will include your full peer review and any attached files.

Reviewer #1: No

Reviewer #2: No

---

## [Author Response · Author response to Decision Letter 0]

9 Feb 2021

Additional requirements:

Please include a copy of Tables 4 and 5 which you refer to in your text on page 13.

Tables 4 and 5:

- Tables 4 and 5 were already in the original paper, and they are also in the revised paper. I have now highlighted the two tables in yellow, in both the document with track changes and in the clean document. Note that these tables appear within the text and not at the end of the paper. 

The tables at the end of the paper (appearing after the references) are tables in the Appendix. They are not the main tables of the paper. I had numbered them as Table A1, Table A2 Table A3 and Table A4. But this seems to have generated confusion, so I have now numbered these Appendix Tables as Tables 6 to 9.

Reviewers:

We have implemented all the suggestions of the reviewers, except those which presume that the study explores mental health before and during COVID. As we explain below, this was not the objective of the study.

Reviewer 1

Major comments

(1) Contribution and related literature: Recent published work has tried to understand how changes in mental wellbeing between pre-COVID and COVID periods can be related to being employed or being furloughed (Banks and Xu, 2020): https://onlinelibrary.wiley.com/doi/full/10.1111/1475-5890.12239. The current paper must frame its methodology, limitations and findings in the context of what has already been done and is already known. How do the methodology and main findings compare with recent COVID studies on employment and mental health? For instance, how do the PHQ-2 and the GHQ-12 measures (the GHQ-12 is used in Banks and Xu, 2020) compare when measuring mental health?

Thank you for this reference, which we now cite in the study. However, in contrast to Banks and Xu (2020), our study does not investigate how mental health changed from the pre-COVID to the COVID periods. Rather, we use the unprecedented job loss that followed the lockdown to explore the implications of job loss for the mental health of a sample of adults who were employed before lockdown. i.e. We use the response to COVID (the lockdown) as a natural experiment, which allows us to overcome the typical problems that arise when exploring the relationship between job loss and mental health viz. that job loss may be endogenous to mental health. This is also the first study that focuses on the implications of job loss for mental health in South Africa, a country with very high rates of unemployment, and where the mental health implications of unemployment are not adequately recognized in public policy. 

In the conclusion, we now recognize that the PHQ-2 is an attractive measure of vulnerability to depression when there are stringent constraints on the length of the instrument, as has been the case for data collection during COVID. But we recognize also that it offers only an initial measure of depressive symptoms, and that in contrast to other measures such as the PHQ-9 and the CES-D 10, it will be less sensitive to variation or severity in mental health.

(2) Selective attrition: Does mental health at baseline predict participation in W1 and W2?

A simple test of attrition using a probit model (Fitzgerald et al. 1998) shows that the probability of not being sampled in NIDS-CRAM Wave 1 is not related to depression scores at baseline (in 2017) (the estimated coefficient = -0.0006 with s.e. = 0.002). The probability of attrition from Wave 1 of NIDS-CRAM to Wave 2 is also not significantly associated with mental health at baseline (coefficient = -0.003, se = 0.005) i.e. There is no evidence of attrition into NIDS-CRAM, or through NIDS-CRAM, based on mental health. We have now recognized this in the section, “Data and methods”. 

(3) Comparability of measures of mental health: What do we know about the relationship between the PHQ-2 and the CES-D 10 from previous studies? What is the fraction of individuals with PHQ>=3 and CES D -10 >=10 in previous studies?

The study explores the association between job loss and depression scores, as measured by PHQ-2. We could not see why the relationship between the PHQ-2 and the CES-D 10 scores would be relevant for the study. In our estimations, we do not directly compare the CES-D 10 scores from 2017 with the PHQ-2 scores from 2020, to draw inferences about how depression scores have changed over time i.e. We are not measuring a change in depression scores from pre-COVID times to COVID times. We also use the scores as a continuous index, rather than imposing thresholds to identify depression. We only include CES-D 10 scores from 2017 as a covariate in the estimations (and therefore estimate lagged models) to provide some control for an unobservable “propensity” to report or experience depressive symptoms. We have now deleted the following sentence from the discussion section, in case this was misleading:

“To the extent that depression scores derived from different instruments can be compared, descriptive analysis identified a higher incidence of depression among the study’s sample in 2020 (24%), compared to 2017 (17%).”

(4) Sample: The study focuses on all adults who were employed in the month before the COVID-lockdown started. Given that the focus is on employment status, it would be more appropriate to focus on a more restricted age group: 18-64. Currently, the maximum age (Table 1) is 89!

The sample includes only those ‘elderly’ who had been employed before lockdown. We chose to analyse the full sample of adults with prior employment for two reasons. At a conceptual level, we did not want to assume that losing employment would not affect the mental health of those who are beyond age 60 (when people are age-eligible for a state pension) or beyond age 64. Many adults who are outside of the working-age range still need to work in South Africa given insufficient savings and retirement benefits. At a practical level, given that the sample was relatively small, we wanted to maximize the sample size for the analysis.

5) Sampling weights: The authors do not mention the use of sampling weights, but they should explain why they do not use the available NIDS (-CRAM) sampling weights. If there is no reason that justifies their choice, I am afraid that sampling weights must be used.

We have given considerable thought to whether the analysis should use the weights. Our concern is that the weights are benchmarked to a sample in 2017, which, as a fifth wave of a panel, was itself not nationally representative. Our sample is also restricted to those who were employed before lockdown, and the weights are not stratified by employment status. We therefore think that a model-based approach is more suited in the context of this study (Winship and Radbill 1994). We have added this explanation in the text and are careful throughout the study to refer only to sample estimates. 

(6) The discussion section can be broken down into two sections: “discussion” and a “conclusion”. Pros and cons of the study must be clearly acknowledged. Cons of the study include the fact that not only mental health is measured on a different scale before the pandemic, but a different collection method is used. This should probably be discussed, if not accounted for.

Thank you for these comments. We have now separated the discussion section as suggested, and we have acknowledged some of the pros and cons of the study in the conclusion. But note that the study does not compare depression scores before the pandemic and during the pandemic, precisely because of the limitations which are highlighted in this comment (that mental health is measured on a different scale and was collected in a different way).

(7) The finding on race/ethnicity (African vs. non-African) is intriguing. Recently, Proto and Quintana-Domeque (2020) show that in the UK there are differences by gender and ethnicity in the deterioration in mental health between pre-COVID and COVID periods. Given the previous research by Proto and Quintana-Domeque (2020) and the findings in this paper, the authors should include the interaction between gender and race. In addition, it is standard practice to define gender as female (and the reference category as male), so that the new regressions should include the following three dummies: Female, African, and Female*African.

Thank you for this suggestion. We have now changed the reference category to male. We re-estimated our models with the interactive dummy (female*African) included, but the results are not significant in any of the estimations and so we have not included the interaction term i.e. There is no evidence that vulnerability to depression during COVID differs by gender and ethnicity. We have recognized this at the end of the discussion section, adding Proto and Quintant-Domeque (2020) to the references. 

Minor comments

a. The title of the manuscript seems a bit misleading: the authors acknowledge that they cannot control for individual fixed effects since they do not observe the dependent variable before the pandemic. This limitation seems important when presenting their evidence as longitudinal.

- The outcome variable is cross-sectional, but we do use longitudinal data in the estimations, by distinguishing employment status in each of the two waves. We also incorporate longitudinal data from 2017 by controlling for depression scores in 2017. The lagged models require panel data. Nonetheless, we have now removed “longitudinal micro-data” from the title.

b. The numbering of tables is not correct. The text refers to Table 3 (p.11), but indeed, there is Table 2 and then “Table 1”, “Table 2” and Table 3” reporting the marginal effects for regressions 1-2, 3 and 4, respectively. Similarly, the text refers to Table 5 (p.13), but there is no Table 5.

- Our apologies, the tables in the appendix were incorrectly numbered, which we have now corrected. The tables within the body of the paper are correctly numbered and complete. 

c. Typos: Table 1 reports a mean of 10.184 for the binary variable “Coronavirus can be avoided”. This should be fixed.

- Thank you for spotting this. We have now corrected the mean value.

d. Pages 9-10: Equations are not numbered. Moreover, the variables y*, x and Y should be indexed with i, and the cut-off mu should not be indexed with i, but with a different letter.

- We have made these changes. 

e. Page 10: The authors write “An insignificant outcome indicates that the assumption has been met.” A more precise statement follows: “An insignificant outcome indicates that there is not enough strong evidence against the parallel regression assumption.”

- Thank-you, we have made this correction.

f. Some of the findings are statistically significant at the 10% level (e.g. Social grants): “The claim that the regression analysis pointed to the importance of social grants (or cash transfers) in providing protection against the incidence of depressive symptoms” seems too strong. The causal language should be tuned down and the authors should not emphasize statistically significant findings at the 10% level.

- We reported on a lower significance threshold in this paper than usual (i.e. the 10% level), given small sample sizes when we disaggregate into sub-groups. But we have moderated the discussion, recognising that associations at the 10% level are only weakly significant. 

References

• Banks, J. and Xu, X. (2020) “The Mental Health Effects of the First Two Months of Lockdown during the COVID-19 Pandemic in the UK,” Fiscal Studies: https://onlinelibrary.wiley.com/doi/full/10.1111/1475-5890.12239

• Proto, E. and Quintana-Domeque, C. (forthcoming) “COVID-19 and mental health deterioration by ethnicity and gender in the UK”, PLOS ONE. Previous version: https://www.iza.org/publications/dp/13503/covid-19-and-mental-health-deterioration-among-bame-groups-in-the-uk

- These are now included in the paper.

Reviewer 2

Reviewer #2: 1. Is the manuscript technically sound, and do the data support the conclusions?

The authors have gone to painstaking lengths to eradicate nearly all concerns about the causal relationship between job loss, furlough, and mental health. Starting with the basics of making a case for an exogenous relationship between job loss and mental health, to noting the sampling to minimize bias found in typically online-only sampling (for rapid sampling, the authors have done an impressive job), to drawing upon a well-established nationally representative survey, to the missingness of data and being transparent that the data appear missing at random based on observed covariates (a scientifically transparent and important distinction from truly missing at random or missing completely at random), to disclosing the inability to account for individual fixed effects pertaining to influences on mental health. I expect that the scholarly community reading this paper may take issue with the exogeneity argument, particularly as those at the margins of the workforce in South Africa—in very tenuous employment, of which many Black South Africans are in—who could have been on the verge of losing their jobs anyway. Nonetheless, this is about as good as one could get for research design and assessing cause and effect. The conclusions drawn are very appropriately rooted in the methods and data.

2. Has the statistical analysis been performed appropriately and rigorously?

In short, yes.

The CES-D 10 is appropriate and externally (and within South Africa among different racial/ethnic groups) validated for measuring depression. The regression models are soundly developed and the authors clearly have not taken any shortcuts—little things like adjusting for age by also including the quadratic are very understated but key to making this research technically sound). Further, noting the parallel regression assumption and how they adjusted for that in their programming is generally considered “best practices” and something that most scholars take for granted/do not discuss when presenting ordered logistic regression models.

I am surprised that the results are not stratified by race/ethnicity as this moderates nearly everything in South Africa. Another suggestion would be to specify your variable coding a bit more. The average reader may be confused by the African versus non-African distinction and what it means in South Africa—I too am a bit confused mainly because I would like to know which category Coloured South Africans fall into (I also recognize that Indian and White South Africans account for only a small share of the overall population). The tables seem out of order or out of place in the manuscript too. The results are all there but the table numbering is off.

- Thank you, we have corrected the tabling numbering.

- Non-African includes Coloureds, Indians and Whites. We have now specified this in the text. We do not distinguish among these groups in the estimations as the coefficients are all individually insignificant (reflecting, at least in part, low sample numbers). 

.

---

## [Decision Letter · Decision Letter 1]

23 Feb 2021

PONE-D-20-38028R1

Job loss and mental health during the COVID-19 lockdown: Evidence from South Africa

PLOS ONE

Dear Dr. Posel,

Thank you for submitting your manuscript to PLOS ONE. After careful consideration, we feel that it has merit but does not fully meet PLOS ONE’s publication criteria as it currently stands. Therefore, we invite you to submit a revised version of the manuscript that addresses the points raised during the review process.

We look forward to receiving your revised manuscript.

Kind regards,

Gabriel A. Picone

Academic Editor

PLOS ONE

Journal Requirements:

Reviewers' comments:

Reviewer's Responses to Questions

**Comments to the Author**

1. If the authors have adequately addressed your comments raised in a previous round of review and you feel that this manuscript is now acceptable for publication, you may indicate that here to bypass the “Comments to the Author” section, enter your conflict of interest statement in the “Confidential to Editor” section, and submit your "Accept" recommendation.

Reviewer #1: (No Response)

Reviewer #2: All comments have been addressed

2. Is the manuscript technically sound, and do the data support the conclusions?

Reviewer #1: Yes

Reviewer #2: (No Response)

3. Has the statistical analysis been performed appropriately and rigorously? 

Reviewer #1: Yes

Reviewer #2: (No Response)

4. Have the authors made all data underlying the findings in their manuscript fully available?

Reviewer #1: Yes

Reviewer #2: (No Response)

5. Is the manuscript presented in an intelligible fashion and written in standard English?

Reviewer #1: Yes

Reviewer #2: (No Response)

6. Review Comments to the Author

Reviewer #1: Report on PONE-D-20-38028R1: "Job loss and mental health during the COVID-19 lockdown: Evidence from South Africa"

Thank you for addressing my previous points.

I think the paper is almost ready for publication. My requested minor revision consists in the following two extensions:

(1) Additional Table 2 for binary dependent variable: Depression (0-1)

The model to be run is:

D(t) = a + bX + cD(t-1) + error term,

where D(t) and D(t-1) are the indicators of depression at t and t-1:

D(t) = 1 if PHQ2 >=3, D(t) = 0 if PHQ2 <3

based on [Kroenke et al. 2003] as discussed on p. 11,

and D(t-1) = 1 if CES-D 10 >=10, D(t-1) = 0 if CES-D 10 < 10

based on [33] as discussed on p. 11.

(2) Additional Table 2 for change in binary dependent variable: Change in Depression

The model to be run is:

D(t) - D(t-1) = a + bX + error term

Required assumption: "Depression scores derived from different instruments (PHQ-2 and CES-D 10) can be compared."

Finally, proofreading is required. One example:

-Kroenke et al (2003) [https://pubmed.ncbi.nlm.nih.gov/14583691/] is missing from the reference list.

-NB. Kroenke et al (2003) on p.11 should be displayed as a [number] not as a name.

Reviewer #2: (No Response)

7. PLOS authors have the option to publish the peer review history of their article (what does this mean?). If published, this will include your full peer review and any attached files.

Reviewer #1: No

Reviewer #2: No

---

## [Author Response · Author response to Decision Letter 1]

7 Mar 2021

As the reviewer suggested, we have estimated the model with a binary dependent variable for depression at time t, and with a binary co-variate of depression in time t-1. Our results remain robust, although some coefficients (e.g. for employment status in period 2) lose significance in some of the regressions.

However, as we explain in the data and methods section of the paper (p.8), the appropriate cut-off for depression in South Africa has been found to vary across the different language groups (as a proxy also for ethnicity) in the country (Baron et al. 2017). Our preferred method therefore is to follow the approach adopted in other studies which have analysed South African micro-data on depression and to use the scales to capture a continuum of depression. (See for example, studies by Ardington and Case 2010; Tomita and Burns 2013; Meffert et al. 2015; Burger et al. 2017.) 

We were also unsure about the motivations for the suggestion to model a change in the likelihood of depression (using the different instruments) as a function of conditions only in time t. Our intention, in including a “baseline” regression score as a covariate, was only to offer some control for a “predisposition” to experience depressive symptoms or to an idiosyncratic over-/under-reporting of depressive symptoms. We therefore attach no weight to a comparison of the different depression scales from 2017 and 2020. We have now been explicit in the methods section (page 7) that robust conclusion about depressive symptoms pre-COVID in 2017 and during COVID in 2020 cannot be drawn.

If the concern is with the inclusion of the CES-D 10 scale as a covariate in the estimations, then we have estimated two further robustness checks:

- First, we estimated the pooled model without the CES-D 10 score as a co-variate.

- Second, we normalized the PHQ-2 scores and the CES-D 10 scores and re-estimated the pooled model.

Our results in these models are all consistent with the results reported in the paper although some of the coefficients in the binary model lose significance. We have included these robustness checks and the binary model in the Appendix (see Tables 10a - c), providing reference to these checks in the results section (pp.13 and 14)

We have also corrected the referencing and proofread the paper more generally.

---

## [Decision Letter · Decision Letter 2]

17 Mar 2021

Job loss and mental health during the COVID-19 lockdown: Evidence from South Africa

PONE-D-20-38028R2

Dear Dr. Posel,

We’re pleased to inform you that your manuscript has been judged scientifically suitable for publication and will be formally accepted for publication once it meets all outstanding technical requirements.

Kind regards,

Gabriel A. Picone

Academic Editor

PLOS ONE

Additional Editor Comments (optional):

Reviewers' comments:

Reviewer's Responses to Questions

**Comments to the Author**

1. If the authors have adequately addressed your comments raised in a previous round of review and you feel that this manuscript is now acceptable for publication, you may indicate that here to bypass the “Comments to the Author” section, enter your conflict of interest statement in the “Confidential to Editor” section, and submit your "Accept" recommendation.

Reviewer #1: All comments have been addressed

2. Is the manuscript technically sound, and do the data support the conclusions?

Reviewer #1: Yes

3. Has the statistical analysis been performed appropriately and rigorously? 

Reviewer #1: Yes

4. Have the authors made all data underlying the findings in their manuscript fully available?

Reviewer #1: Yes

5. Is the manuscript presented in an intelligible fashion and written in standard English?

Reviewer #1: Yes

6. Review Comments to the Author

Reviewer #1: Thank you for addressing all my comments.

Last thing: make sure you proofread the article one more time.

In equation (2), "y*" should be replaced with "if y*i" and some of the weak inequalities should be replaced with strict inequalities (e.g. replacing <= with <)

7. PLOS authors have the option to publish the peer review history of their article (what does this mean?). If published, this will include your full peer review and any attached files.

Reviewer #1: No

---

## [Editor Report · Acceptance letter]

22 Mar 2021

PONE-D-20-38028R2 

Job loss and mental health during the COVID-19 lockdown: Evidence from South Africa  

Dear Dr. Posel:

I'm pleased to inform you that your manuscript has been deemed suitable for publication in PLOS ONE. Congratulations! Your manuscript is now with our production department. 

Kind regards, 

on behalf of

Dr. Gabriel A. Picone 

Academic Editor

PLOS ONE